OBSERVATION

# Detecting Waning Serological Response with Commercial Immunoassays: 18-Month Longitudinal Follow-up of Anti-SARS-CoV-2 Nucleocapsid Antibodies

Yu Nakagama,[a,b] Yuko Komase,[c] Natsuko Kaku,[a,b] Yuko Nitahara,[a,b] Evariste Tshibangu-Kabamba,[a,b] Tomoyo Tominaga,[d] Hiroko Tanaka,[d] Tomoaki Yokoya,[d] Minako Hosokawa,[d] Yasutoshi Kido[a,b]

[a]Department of Parasitology, Graduate School of Medicine, Osaka City University, Osaka, Japan
[b]Research Center for Infectious Disease Sciences, Graduate School of Medicine, Osaka City University, Osaka, Japan
[c]Department of Respiratory Internal Medicine, St. Marianna University School of Medicine, Yokohama-city Seibu Hospital, Yokohama, Japan
[d]Department of Health Management, St. Marianna University School of Medicine, Yokohama-city Seibu Hospital, Yokohama, Japan

**ABSTRACT** Past severe acute respiratory syndrome coronavirus 2 (SARS-CoV-2) infection is an important determinant of protection from reinfection and of postvaccine immune responses. Herein, we conducted a follow-up analysis of health care workers previously infected with coronavirus disease 2019 (COVID-19) with the aim of evaluating different immunoassays for their capability in detecting the waning anti-SARS-CoV-2 immune responses and accuracy in documenting past SARS-CoV-2 infections. We evaluated serum antinucleocapsid antibody levels in convalescent individuals following a 1.5-year interval from SARS-CoV-2 infection. Three different commercial immunoassays that qualitatively measure serum antibodies targeting the SARS-CoV-2 nucleocapsid protein, namely, the Abbott Architect SARS-CoV-2 IgG, the Euroimmun anti-SARS-CoV-2 NCP enzyme-linked immunosorbent assay (ELISA) IgG, and the Roche Elecsys anti-SARS-CoV-2, were tested for comparison of detectability. A total of 38 individuals consented to participating in this follow-up analysis. From assay to assay, seropositivity rate at 18 months from infection varied from lowest at 42% to highest at 92%. The Roche Elecsys immunoassay, dependent on the dual-antigen antibody detection method and tuned for the detection of high avidity antibodies, was most capable of accurately documenting past SARS-CoV-2 infections. Different immunoassays showed variable capability of determining previous infection status under waning antibody concentrations. Immunoassays with lower detection limits are to be selected, and adjusted thresholds are to be considered in order to maximize the tests' performance.

**IMPORTANCE** Past SARS-CoV-2 infection is an important determinant of protection from reinfection and of postvaccine immune responses. Our results show that different immunoassays, by design, harbor variable capability of tracking SARS-CoV-2 infection under waning antibody concentrations. With each recovered patient standing at a unique time point along the decline curve of antibodies, precise estimation of COVID-19 cumulative incidence remains a challenge. Since future surveillance studies will be targeting more than ever heterogenous cohorts, selecting the appropriate immunoassay is crucial in order to assure reliable decisions about an individual's previous infection status.

**KEYWORDS** COVID-19, SARS-CoV-2, waning antibody, serological kinetics

The waning antibody levels following recovery from natural severe acute respiratory syndrome coronavirus 2 (SARS-CoV-2) infection complicate the precise estimation of seropositivity rates within societies. Nevertheless, seropositivity is an important determinant of an individual's protection level against reinfection, and seroprevalence is a guide to understanding how far societies are from attaining herd immunity. Past SARS-CoV-2 infection is also a strong predictor of postvaccine immune responses (1). Individuals with a

Address correspondence to Yasutoshi Kido, kido.yasutoshi@med.osaka-cu.ac.jp.

The authors declare a conflict of interest. This research was supported by Japan Agency for Medical Research and Development [JP20jk0110021] and Osaka City University Strategic Research Grant [OCU-SRG2021_YR09]. Yu Nakagama and Yasutoshi Kido receive financial support from Abbott Japan LLC, Japan, outside the work.

10.1128/spectrum.00986-22   **1**

previous SARS-CoV-2 infection not only enjoy longer-lasting postvaccination antibody levels but also are known to achieve more efficient protection from clinical reinfection (2). For those with pauci-/asymptomatic infections who, for various reasons, did not receive an acute diagnosis, its later confirmation is only possible serologically (3, 4). Therefore, robust serological assays are increasingly needed that assure reliable results despite waning antibody responses. Herein, we conducted a follow-up analysis of serological status among a previously characterized cohort of coronavirus disease 2019 (COVID-19) convalescent individuals (3). Our aim was to evaluate different immunoassays for their capability in detecting waning anti-SARS-CoV-2 immune responses.

This study was a follow-up analysis of the serum antibody level among a cohort of health care workers initially infected during the first wave of the pandemic in Japan during April and May 2020 (3). In the preceding study, 64 of 414 health care workers from a tertiary care hospital were affected by a COVID-19 outbreak. Cases were well-described regarding their infection status both molecularly and serologically. During the third and fourth week of November 2021, approximately 1.5 years after the initial assessment of serological status, the same cohort was inspected for serum SARS-CoV-2 antinucleocapsid antibody level. Each individual donated a serum sample after giving written consent for participation. The study was approved by the Osaka City University Institutional Ethics Committee (number 2020-003). Serum samples were tested using three different commercial immunoassays, which qualitatively measure serum antibodies targeting the SARS-CoV-2 nucleocapsid protein. For the Abbott Architect SARS-CoV-2 IgG (Abbott Laboratories, Chicago, IL, USA) and the Euroimmun anti-SARS-CoV-2 NCP enzyme-linked immunosorbent assay (ELISA) IgG (Euroimmun Medizinische Labordiagnostika AG, Lübeck, Germany) immunoassays, adjusted thresholds of 0.8 (index sample/control [S/C]) and 0.8 (ratio sample/calibrator [S/C]), respectively, were used to define positivity. This approach has been efficient in maximizing test sensitivity while maintaining specificity compared with the manufacturers' predefined assay thresholds of 1.0 (index S/C) and 1.1 (ratio S/C) (5–7). The Roche Elecsys anti-SARS-CoV-2 (Roche Diagnostics, Rotkreuz, Switzerland) immunoassay results were judged based on the threshold of 1.0 (cutoff index [COI]), originally proposed by the manufacturer. Antibody levels were expressed in geometric means with 95% confidence intervals (CI). Comparisons of antibody levels were made, per assay, between the early 2-month convalescent and the late 18-month convalescent phases by Wilcoxon matched-pairs signed-rank test.

Out of 64 individuals from the previously described health care worker cohort (median age, 36; range, 23 to 62) with well-defined seropositivity against SARS-CoV-2, a total of 38 consented to participate in this follow-up analysis. Antinucleocapsid antibodies had been detected in 38/38 (100%), 37/38 (97%), and 38/38 (100%) participants at 2 months postinfection using the Abbott Architect, the Euroimmun ELISA, and the Roche Elecsys immunoassays, respectively. At 18 months postinfection, the same assays returned persistent serological evidence of previous infection in 16/38 (42%), 18/38 (47%), and 35/38 (92%), respectively. The geometric means (95% CI) of antinucleocapsid antibodies have generally declined (Abbott Architect, 5.00 [4.20 to 5.95] versus 0.60 [0.42 to 0.86] [index S/C]; Euroimmun ELISA, 3.40 [2.80 to 4.13] versus 0.77 [0.62 to 0.97] [ratio S/C]; Roche Elecsys, 77.0 [56.4 to 105] versus 22.2 [13.1 to 37.9] [COI]) along the time course after recovery ($P < 0.0001$), with only a few exceptional cases that have shown a rise in their antibody level measured with the Roche Elecsys assay (Fig. 1A to C). Agreement between measured antibody levels were analyzed across the commercial immunoassays. The measured antibody levels from the Roche Elecsys immunoassay showed moderate to strong correlation with the results from the other two immunoassay platforms. Interestingly, correlation became stronger at the later 18-month convalescent phase, with Pearson's correlation coefficients raising from an $r$ of 0.70 (Abbott Architect at 2 months) and an $r$ of 0.63 (Euroimmun ELISA at 2 months) to an $r$ of 0.90 (Abbott Architect at 18 months) and an $r$ of 0.87 (Euroimmun ELISA at 18 months) (Fig. 2A and B).

Few studies have described the kinetics of anti-SARS-CoV-2 antibodies extending beyond 1 year of follow-up after a natural infection (8). Up to 7.5 months, the Roche Elecsys antinucleocapsid immunoassay has detected a sustained antibody response in convalescent individuals (9). Conversely, a study has found a seroreversion (loss of antibodies) rate of over 30% at 8 months of follow-up with the Abbott Architect antinucleocapsid immunoassay

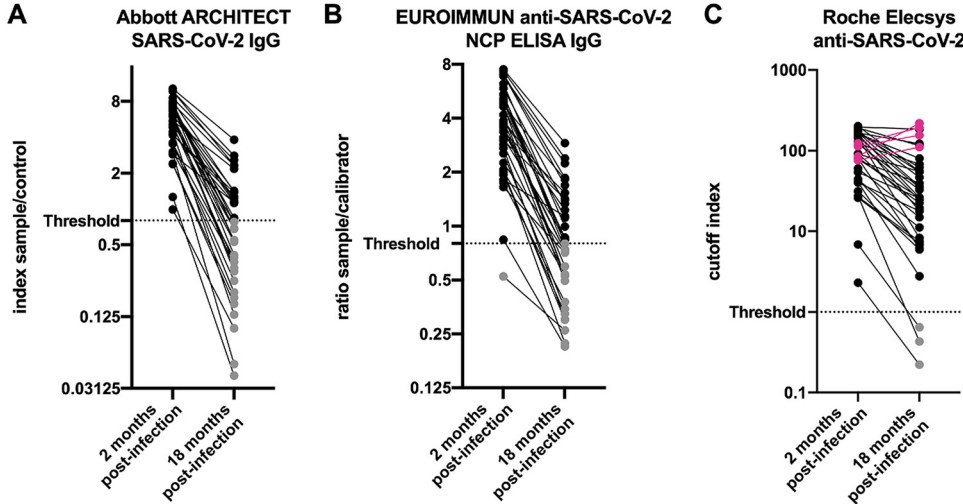

**FIG 1** Serum antinucleocapsid antibody level of participants previously infected with COVID-19 measured at 2 and 18 months postinfection with the Abbott Architect (A), Euroimmun ELISA (B), and Roche Elecsys (C) immunoassays. Horizontal dotted line shows the positivity threshold of the assay. Antibody levels above and below the assay threshold are indicated in black and gray symbols, respectively. Individuals who showed raising antibody levels are indicated in magenta.

(10). Our results, with one of the longest-term follow-up collection of data, further build on the growing evidence that the interpretation of serological kinetics against SARS-CoV-2 are dependent on assay platform and further stress the importance of selecting the appropriate assay that will assure reliable results (11). It was interesting to have observed better correlations among the three tested immunoassays at the later 18-month convalescent phase when serum antibodies were rich in mature, highly avid antibodies. We have reported that the uniqueness in design of the Roche Elecsys immunoassay, namely, the dual-antigen binding method, makes the assay less likely to capture low-affinity antibodies that are predominant during the early stage of convalescence (12). Thus, when assessing serum samples from early-convalescence, the different immunoassays' variable likelihood of detecting such low-affinity antibodies may have a larger impact on the measured antibody level and may subsequently lead to poorer interassay agreement. In later convalescence, the affinity maturation of antibodies progresses and the serum samples become rich in higher-affinity antibodies secreted by the long-lived plasma cells. These higher-affinity antibodies are more readily detected with the dual-antigen binding method. Therefore, when measured with the Roche Elecsys immunoassay, an individual with highly efficient maturation in the affinity of antibodies may experience an increase in relative antibody levels over time (12) as observed in several of the participants from the present study. The same logic appears to apply to the vaccine-elicited antibodies, where interassay agreement of antibody level remained poor during the earlier stage of postvaccination, and an increase over time in Roche Elecsys antibody level was observed in some significant proportion of vaccinees (13).

Antinucleocapsid serology, assessed in parallel with neutralizing antibody responses, shall guide us to improved understanding of our hybrid immunity against SARS-CoV-2. Indeed, no policies currently consider an individual's prior infection status nor serology in the designing of vaccination strategies. However, faced with increasing evidence that vaccinees with a previous SARS-CoV-2 infection are better protected from the virus than those without (14, 15), the additional boosters' risk-benefit profiles for the previously infected individuals, as well as their optimal timing and dose, shall stay a matter of continuing interest.

Limitations of the study are mentioned below. Firstly, the clinical performance of an immunoassay will highly depend on the analyzed cohort. The present study has targeted a limited number of individuals, health care workers mostly of a young age with few comorbidities, resulting overall in rather mild COVID-19 phenotypes. The observed immune response may not be representative of the overall population. Being underpowered in design has also hindered the study from assessing for any predictors of seroreversion. However, as we and

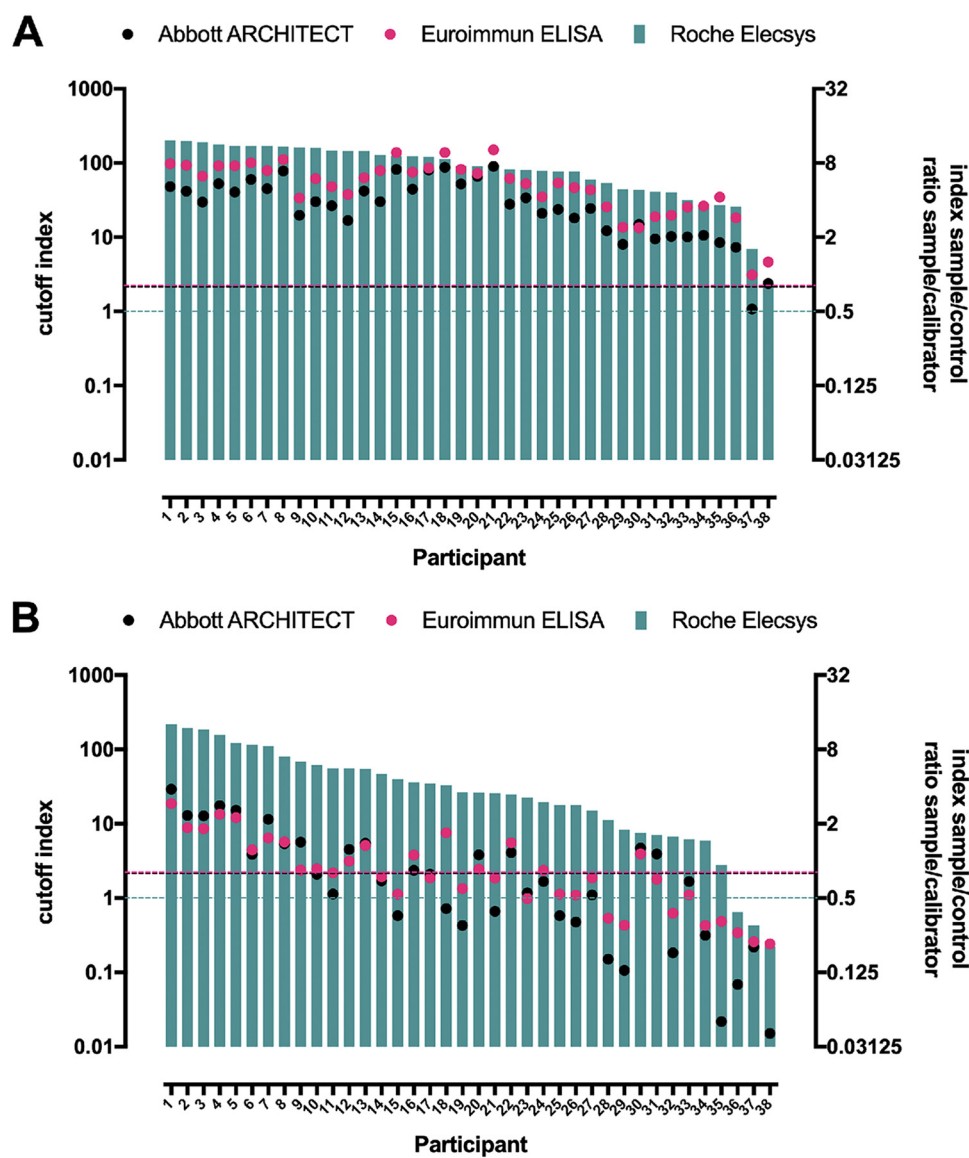

**FIG 2** Correlation between antinucleocapsid antibody levels measured with different assay platforms at 2 months (A) and at 18 months (B) postinfection. Antibody levels from the participants at each time point were arranged in descending order. Horizontal dotted line shows the positivity threshold of the assay indicated in the same color.

others have previously described, COVID-19 severity is among the strongest determinant of an individual's serological response (3, 16). Larger scaled analyses are warranted to further explore the association between symptom severity and seroreversion rates. Secondly, having not been actively screened for SARS-CoV-2 infection during the follow-up interval, participants are not fully denied of additional viral exposures. As health care workers attending a teaching hospital, however, they were intensely monitored for their health status and reinfections were never documented.

In conclusion, different immunoassays showed variable capability in determining previous infection status under waning antibody concentrations. Immunoassays with lower detection limits are to be selected, and adjusted thresholds are to be considered in order to maximize test performance.

## ACKNOWLEDGMENTS

We are grateful to all participants of the study. We thank Sachie Nakagama from the Department of Parasitology, Graduate School of Medicine, Osaka City University for technical assistance in measuring antibodies.

This research was supported by Japan Agency for Medical Research and Development (JP20jk0110021, JP20he1122001, and JP20wm0125003), Osaka City University Strategic Research Grant (OCU-SRG2021_YR09), and the Osaka City University Special Reserves Fund for COVID-19.

Yu Nakagama and Yasutoshi Kido receive financial support from Abbott Japan LLC, Japan, outside of this work.

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
