## [Reviewer comments · Microbiology Spectrum]

Microbiology Spectrum

Detecting the waning serological response with commercial immunoassays; 18-month longitudinal follow-up of anti-SARS-CoV-2 nucleocapsid antibodies.

Yu Nakagama, Yuko Komase, Natsuko Kaku, Yuko Nitahara, Evariste Tshibangu-Kabamba, Tomoyo Tominaga, Hiroko Tanaka, Tomoaki Yokoya, Minako Hosokawa, and Yasutoshi Kido

Corresponding Author(s): Yasutoshi Kido, Osaka City University

Review Timeline:

Submission Date:	March 17, 2022
Editorial Decision:	May 11, 2022
Revision Received:	June 22, 2022
Accepted:	June 28, 2022

Editor: Eleanor Powell

Reviewer(s): The reviewers have opted to remain anonymous.

Transaction Report:

DOI: <https://doi.org/10.1128/spectrum.00986-22>

May 11, 2022

Dr. Yasutoshi Kido
Osaka City University
Dept. of Parasitology and Research Center for Infectious Disease Sciences
Asahimachi1-4-3
Abeno-ku
Osaka 5458585
Japan

Re: Spectrum00986-22 (Detecting the waning serological response with commercial immunoassays; 18-month longitudinal follow-up of anti-SARS-CoV-2 nucleocapsid antibodies.)

Dear Dr. Yasutoshi Kido:

Thank you for submitting your manuscript to Microbiology Spectrum. After receiving feedback from reviewers, modifications are necessary before potential publication.

Link Not Available

Sincerely,

Eleanor Powell

Journals Department
Reviewer comments:

Reviewer #1 (Comments for the Author):

The authors nicely describe as an Observation the longitudinal study of previously infected healthcare workers with COVID-19 to evaluate different immunoassays for detecting waning anti-SARS-CoV-2 immune responses. A value of this study is that it provides diagnostic evaluations for 3 different commercially available immunoassays with follow-up analysis of seropositivity out

to 18 months post-infection showing the difference in titers as a reflection of assay performance at this time point. As we are continuing to learn about the immunity to SARS-CoV-2 from primary infections and vaccines this observation is important to the scientific community.

Major comments:

Recent studies have shown that IgG levels can be impacted by the type of infection the patient originally experienced (symptomatic vs. asymptomatic) (Huynen et al 2022, Gerhards et al 2021). Huynen et al. 2022 showed IgG levels that were higher and increased over time in symptomatic subjects but were lower and stable in asymptomatic subjects. Can the authors comment on whether there was any impact on the IgG levels in the participants of this study based on the earlier infection and symptoms?

Line 240 and 247: Should state 18-months

Minor Comments:

Line 78: Define COVID-19 here as opposed to line 84.

Line 92: Add manufacturer information for each immunoassay

Line 94: How do the adjusted thresholds compare to the instructions for use from the manufacturer, please address.

Line 113: Address why the titers for these patients may have increased.

Line 127: Remove the redundant sentence "The pandemic having..." and consider revising it in line 61.

Line 135: Consider adding the sentence: "Our results..." to the Importance section as it highlights an important aspect of this Observation.

Reviewer #2 (Comments for the Author):

This study compared the ability of three different high throughput assays to detect anti-nucleocapsid antibodies 18 months after initial infection. They show notable differences in positivity rates for the Roche assay as compared to Abbott and Euroimmun.

Comments for the authors:

1. Indicate that these are qualitative assays against the nucleocapsid antigen in the abstract and methods section.
2. Clarify that there really are no policies in place that currently take into account serologic testing as a means to determine the need for additional boosters.
3. One of the key limitations here is the limited number of individuals included in this long term study.
4. Can the authors comment on how their findings should be used to guide the public health response, if at all? This would circle back to their initial comments on use of serology to help potentially guide vaccination needs in the future.

Staff Comments:

Preparing Revision Guidelines

Please return the manuscript within 60 days; if you cannot complete the modification within this time period, please contact me. If you do not wish to modify the manuscript and prefer to submit it to another journal, please notify me of your decision immediately so that the manuscript may be formally withdrawn from consideration by Microbiology Spectrum.

Responses to comments made by Reviewer #1:

The authors nicely describe as an **Observation** the longitudinal study of previously infected healthcare workers with COVID-19 to evaluate different immunoassays for detecting waning anti-SARS-CoV-2 immune responses. A value of this study is that it provides diagnostic evaluations for 3 different commercially available immunoassays with follow-up analysis of seropositivity out to 18 months post-infection showing the difference in titers as a reflection of assay performance at this time point. As we are continuing to learn about the immunity to SARS-CoV-2 from primary infections and vaccines this observation is important to the scientific community.

Comment 1;

Recent studies have shown that IgG levels can be impacted by the type of infection the patient originally experienced (symptomatic vs. asymptomatic) (Huynen et al 2022, Gerhards et al 2021). Huynen et al. 2022 showed IgG levels that were higher and increased over time in symptomatic subjects but were lower and stable in asymptomatic subjects. Can the authors comment on whether there was any impact on the IgG levels in the participants of this study based on the earlier infection and symptoms?

We honestly thank the reviewer for the very interesting comment and perfectly agree that the severity of COVID-19 is one of the strongest determinants of an individual's immune response, and may as well have an effect on future seroreversion rates.

Along with the works by Huynen and Gerhards, our group has also previously stated in an article already published in your journal that pauci-/asymptomatic COVID-19 cases are associated with attenuated immune responses (*Nakagama Y, et al. Microbiol Spectr. 2021 Oct 31;9(2):e0108221. doi: 10.1128/Spectrum.01082-21*). Combining the evidence, symptomatic cases are likely to elicit a more robust, as well as a longer-lasting, immune responses compared with asymptomatic infections. In the present study, however, we were too underpowered to assess for any correlation between COVID-19 severity and seroreversion rates. We have listed this among the *Limitations* and stated the need for a larger-scaled analysis to efficiently approach this

hypothesis.

Firstly, [...]. Being underpowered in design has also hindered the study from assessing for any predictors of seroreversion. However, as we and others have previously described, COVID-19 severity is among the strongest determinants of an individual's serological response(3,16). Larger scaled analyses are warranted to further explore the association between symptom severity and seroreversion rates.

page 10, lines 233–241

A reference has been added.

page 15, lines 365–368

16. Huynen P, Grégoire C, Gofflot S, Seidel L, Maes N, Vranken L, Delcour S, Moutschen M, Hayette MP, Kolh P, Melin P, Beguin Y. 2022. Long-term longitudinal evaluation of the prevalence of SARS-CoV-2 antibodies in healthcare and university workers. *Sci Rep* 12:5156.

A further interesting aspect regarding your comment was the possible contrast between severity group in the kinetics of antibody response. We have clearly shown in a study published in your sister journal (Ref #12, *Nakagama Y, et al. 2022. A dual-antigen SARS-CoV-2 serological assay reflects antibody avidity. J Clin Microbiol* 60:e0226221.) that the immunoassays' design is another major determinant of antibody kinetics.

Immunoassays, each to different extent, reflect the maturity and avidity of the serum antibodies. The design of the Roche Elecsys immunoassay, namely the dual-antigen binding method, makes the assay less likely to capture low-affinity antibodies that are predominant during the early stage of convalescence. Thus, when assessing sera from early-convalescence, the different immunoassays' variable likelihood of detecting such low-affinity antibodies has larger impact on measured antibody level, and subsequently lead to poorer inter-assay agreement. In later-convalescence, the sera become rich in higher-affinity antibodies which are more

readily detected with the dual-antigen binding method. Therefore, when measured with the Roche Elecsys immunoassay, an individual having experienced highly efficient affinity maturation of their antibodies show an increase in relative antibody levels over time (Ref #12, Nakagama Y, et al. 2022. *A dual-antigen SARS-CoV-2 serological assay reflects antibody avidity. J Clin Microbiol* 60:e0226221.). Further, we have recently shown in a different study that the logic is generalizable and applies also to the vaccine-elicited antibodies (Ref #13, Matsuura T, Fukushima W, Nakagama Y, et al. 2022. *Kinetics of Anti-SARS-CoV-2 Antibody Titer in Healthy Adults Up to 6 Months after BNT162b2 Vaccination Measured by Two Immunoassays: A Prospective Cohort Study in Japan. JVac-D-22-00871, Available at SSRN: <https://ssrn.com/abstract=4107500>*).

We have realized that the *Discussions* section has been somewhat redundant (e.g. rephrasing of what has already been said in the *Introduction*) in the initial submission. Therefore, in the revised version, we decided to put more weight on the aforementioned discussions, and the text appears now as below.

We have ~~recently~~ reported that ~~immunoassays [...] the analyte serum antibodies~~the uniqueness in design of the Roche Elecsys immunoassay, namely the dual-antigen binding method, makes the assay less likely to capture low-affinity antibodies that are predominant during the early stage of convalescence(12). ~~The design of [...] the receiver operating characteristic curve analysis~~Thus, when assessing sera from early-convalescence, the different immunoassays' variable likelihood of detecting such low-affinity antibodies may have larger impact on the measured antibody level, and subsequently lead to poorer inter-assay agreement. In later-convalescence, the affinity maturation of antibodies progresses and the sera become rich in higher-affinity antibodies secreted by the long-lived plasma cells. These higher-affinity antibodies are more readily detected with the dual-antigen binding method. Therefore, when measured with the Roche Elecsys immunoassay, an individual with highly efficient maturation in the affinity of antibodies may experience an increase in relative antibody levels over time(12), as observed in several of the participants from the present study. The same logic appears to apply also to the vaccine-elicited antibodies, where inter-assay agreement of antibody level remained poor during the earlier-stage of post-vaccination and an increase over time in Roche Elecsys antibody level was observed in some significant proportion of

page 8, lines 173–208

Comment 2;

Line 240 and 247: Should state 18-months

We honestly thank the reviewer for the correction, and apologize about the erroneous description in the initial submission. The correction appears now, as below.

Figure 1. Serum anti-nucleocapsid antibody level of participants previously infected with COVID-19 measured at 2- and 18-months post-infection, with (A) the Abbott Architect, (B) EUROIMMUN ELISA, and (C) Roche Elecsys immunoassays. Horizontal dotted line shows the positivity threshold of the assay. Antibody levels above and below the assay threshold are indicated in black and grey symbols, respectively. Individuals who showed raising antibody levels are indicated in magenta.

Figure 2. Correlation between anti-nucleocapsid antibody levels measured with

page 16, lines 381–389

Comment 3;

Line 78: Define COVID-19 here as opposed to line 84.

Thank you, reviewer, for the correction. We also apologize about the erroneous description in the initial submission. The correction appears now, as below.

We herein conducted a follow-up analysis of the serological status among a previously characterized cohort of coronavirus disease 2019 (COVID-19) convalescent individuals(3). [...].

This study was a follow-up analysis of the serum antibody level among a cohort of healthcare workers initially infected during the first wave of the pandemic in Japan during April–May 2020(3). In the preceding study, 64 of 414 healthcare workers from a tertiary care hospital were affected by a ~~coronavirus disease 2019~~ (COVID-19)-outbreak.

page 5, line 92–108

Comment 4;

Line 92: Add manufacturer information for each immunoassay

We thank the reviewer for the comment. We have added some information to the original manuscript. I believe that the information below, correctly specifies the

manufacturers.

For the Abbott ARCHITECT SARS-CoV-2 IgG (Abbott **Laboratories**, Illinois, USA) and the EUROIMMUN anti-SARS-CoV-2 NCP ELISA IgG (EUROIMMUN **Medizinische Labordiagnostika AG**, Lübeck, Germany) immunoassays, adjusted thresholds of 0.8 [index Sample/Control (S/C)] and 0.8 [ratio Sample/Calibrator (S/C)], respectively, were used [...] The Roche Elecsys anti-SARS-CoV-2 (Roche **Diagnostics, Rotkreuz**, Switzerland) immunoassay results were judged based on the threshold of 1.0 [cutoff index (COI)], originally proposed by the manufacturer.

page 6, lines 115–124

Comment 5;

Line 94: How do the adjusted thresholds compare to the instructions for use from the manufacturer, please address.

We thank the reviewer for the comment. Our adjusted thresholds are now presented in comparison with the original manufacturers' cutoff values.

page 6, lines 118–122

[...] adjusted thresholds of 0.8 [index Sample/Control (S/C)] and 0.8 [ratio Sample/Calibrator (S/C)], respectively, were used to define positivity ~~with the aim of~~. **This approach has been efficient in** maximizing test sensitivity while maintaining specificity, **as compared with the manufacturers' pre-defined assay thresholds of 1.0 [index S/C] and 1.1 [ratio S/C] (5–7).**

Comment 6;

Line 113: Address why the titers for these patients may have increased.

We thank the reviewer for the comment and noticed that we have fell short of sufficiently explaining the mechanism.

As stated in our *'Response to Comment #1'*, the design of the Roche Elecsys immunoassay, namely the dual-antigen binding method, makes the assay less likely to

capture low-affinity antibodies that are predominant during the early stage of convalescence. In later-convalescence, the sera become rich in higher-affinity antibodies which are more readily detected with the dual-antigen binding method. Therefore, when measured with the Roche Elecsys immunoassay, an individual having experienced highly efficient affinity maturation of their antibodies may show an increase in relative antibody levels over time. Further, we have recently shown in a different study that the logic is generalizable and applies also to the vaccine-elicited antibodies. The findings have been added to the discussion, with relevant citations. Ref #13 has been replaced so as to better fit the context.

page 8, lines 173–208

We have reported that ~~immunoassays [...] the analyte serum antibodies~~the uniqueness in design of the Roche Elecsys immunoassay, namely the dual-antigen binding method, makes the assay less likely to capture low-affinity antibodies that are predominant during the early stage of convalescence(12). ~~The design of [...] the receiver operating characteristic curve analysis~~Thus, when assessing sera from early-convalescence, the different immunoassays' variable likelihood of detecting such low-affinity antibodies may have larger impact on the measured antibody level, and subsequently lead to poorer inter-assay agreement. In later-convalescence, the affinity maturation of antibodies progresses and the sera become rich in higher-affinity antibodies secreted by the long-lived plasma cells. These higher-affinity antibodies are more readily detected with the dual-antigen binding method. Therefore, when measured with the Roche Elecsys immunoassay, an individual with highly efficient maturation in the affinity of antibodies may experience an increase in relative antibody levels over time(12), as observed in several of the participants from the present study. The same logic appears to apply also to the vaccine-elicited antibodies, where inter-assay agreement of antibody level remained poor during the earlier-stage of post-vaccination and an increase over time in Roche Elecsys antibody level was observed in some significant proportion of

page 14, lines 331–357

~~13. National SARS-CoV-2 Serology Assay Evaluation Group. 2020. Performance characteristics of five immunoassays for SARS-CoV-2: a head-to-head benchmark comparison. Lancet Infect Dis 20:1390–1400.~~Matsuura T, Fukushima W, Nakagama Y, Kido Y, Kase T, Kondo K, Kaku N, Matsumoto K, Suita A, Komiya E, Mukai E, Nitahara Y, Konishi A, Kasamatsu A, Nakagami-Yamaguchi E, Ohfuji S, Kaneko Y, Kaneko A, Kakeya H, Hirota Y. 2022. Kinetics of Anti-SARS-CoV-2 Antibody Titer in Healthy Adults Up to 6 Months after BNT162b2 Vaccination Measured by Two Immunoassays: A Prospective Cohort Study in Japan. JVAC-D-22-00871, Available at SSRN: <https://ssrn.com/abstract=4107500>.

Comment 7;

Line 127: Remove the redundant sentence "The pandemic having..." and consider revising it in line 61.

We thank the reviewer for the comment, and apologize for our redundancy. The sentence has now been discarded, and the context had it appeared is now somewhat rephrased, as below.

page 4, lines 65–69

~~The pandemic having prevailed for more than two years,~~With each recovered patient ~~will now positioning~~standing at a unique time point along the decline curve of antibodies, ~~precise estimation of COVID-19 cumulative incidence remains a challenge.~~ ~~Thus,~~Since future surveillance studies will be targeting more than ever heterogenous cohorts, selecting the appropriate immunoassay is crucial in order to assure reliable decision on an individual's previous infection status.

Conversely, a study has found a seroreversion (loss of antibodies) rate of over 30% at eight months of follow-up with the Abbott ARCHITECT anti-nucleocapsid immunoassay(10).~~The pandemic having prevailed for more than two years, each recovered patient will now position at a unique time point along their decline curve of antibodies. [...] Since past COVID-19 infection is a strong predictor of protection from symptomatic reinfection and of post vaccine immune responses, it is considerably applicable to prioritize the distribution of primary and further booster vaccinations based on anti-nucleocapsid antibody serology (11,12).~~ Our results, with one of the longest-term follow-up data, further build on the growing evidence that the interpretation of serological kinetics against SARS-CoV-2 are dependent on assay platform, and further stress the importance of selecting the appropriate assay that will assure reliable results(11).

page 8, lines 166–171

Comment 8;

Line 135: Consider adding the sentence: "Our results..." to the Importance section as it highlights an important aspect of this Observation.

Thank you reviewer for the suggestion. The section has been reformatted as suggested.

page 4, lines 63–65

Our results show that different immunoassays, by design, harbor variable capability of tracking SARS-CoV-2 infection under waning antibody concentrations.

Responses to comments made by Reviewer #2:

This study compared the ability of three different high throughput assays to detect anti-nucleocapsid antibodies 18 months after initial infection. They show notable differences in positivity rates for the Roche assay as compared to Abbott and Euroimmun.

Comment 1;

Indicate that these are qualitative assays against the nucleocapsid antigen in the abstract and methods section.

We appreciate the comment. The qualitative nature of the assays are now clearly indicated in the *Abstract* and the *Methods* sections. Taking this comment into consideration, we have replaced the term “titer”, which gives the impression of a strictly ‘quantitative’ measure, with the word “level”, wherever appropriate throughout the text. Every substitution of the term is now highlighted in red font using the Word track-changes function.

page 4, lines 45–48

Three different commercial immunoassays **which qualitatively measure serum antibodies targeting the SARS-CoV-2 nucleocapsid protein**, namely [...], were tested for comparison of detectability.

page 6, lines 114–115

Serum samples were tested ~~for antibodies targeting the SARS-CoV-2 nucleocapsid protein~~ **using three different commercial immunoassays which qualitatively measure serum antibodies targeting the SARS-CoV-2 nucleocapsid protein.**

Comment 2;

Clarify that there really are no policies in place that currently take into account serologic testing as a means to determine the need for additional boosters.

We fully appreciate the reviewer’s comment. In fact, it is absolutely true that no policies currently consider an individual's prior infection status nor serology in the

designing of vaccination strategies. Considering the still controversial issues regarding the future needs of additional boosters, as well as their optimal timing and dose, in previously infected individuals, we modified the text so as to avoid declarative phrasings.

We have modified the discussions as below.

~~Aside from surveillance purposes, previous infection status may serve as a potential index to differentiate still vulnerable naïve populations from seroprotected individuals. Since past COVID-19 infection is a strong predictor of protection from symptomatic reinfection and of post-vaccine immune responses, it is considerably applicable to prioritize the distribution of primary and further booster vaccinations based on anti-nucleocapsid antibody serology (11,12).~~

page 8, lines 168–

page 9, lines 209–232

Anti-nucleocapsid serology, assessed in parallel with the neutralizing antibody responses, shall guide us to improved understanding of our hybrid immunity against SARS-CoV-2. Indeed, no policies currently consider an individual's prior infection status nor serology in the designing of vaccination strategies. However, faced with the increasing evidence that vaccinees with a previous SARS-CoV-2 infection are better protected from the virus than those without(13,14), the additional boosters' risk-benefit profiles for the previously infected individuals, as well as their optimal timing and dose, shall stay a matter of continuing interest.

Comment 3;

One of the key limitations here is the limited number of individuals included in this long term study.

We fully appreciate the reviewer's comment. We have now clearly mentioned this in the Limitations section.

The present study has targeted ~~a limited number of individuals~~: healthcare workers ~~mostly generally~~ of younger age with few comorbidities, resulting overall in rather mild COVID-19 phenotypes; ~~only 1/38 required oxygen supplementation~~. The immune response ~~may not be representative of the overall population may differ in kinetics and intensity if the analyzed cohort are to include cases of greater severity~~.

Comment 4;

Can the authors comment on how their findings should be used to guide the public health response, if at all? This would circle back to their initial comments on use of serology to help potentially guide vaccination needs in the future.

We appreciate the reviewer's comment. The incremental effect of prior COVID-19 infection on post-vaccine immune protection (hybrid immunity) has been increasingly evaluated, as stated in the WHO's interim statement (<https://www.who.int/news/item/01-06-2022-interim-statement-on-hybrid-immunity-and-increasing-population-seroprevalence-rates>). For those with prior infection, the decay in antibody titer following a set of SARS-CoV-2 vaccination was far more gradual, leading to an approximate 4-fold change in the long-term antibody titer (*Havervall S, et al. Clin Transl Immunology. 2022 Apr 18;11(4):e1388. doi: 10.1002/cti2.1388*). Analysis from the UK's national COVID-19 Survey has also estimated that, after two BNT162b2 doses, 67%-efficacy protective antibody titer shall last longer for those following a natural infection; 5–8 months for naïve vs 1–2 years for those with prior infection (*Wei J, et al. Nat Med. 2022 May;28(5):1072-1082. doi: 10.1038/s41591-022-01721-6*).

We believe that, with more evidence, integrating infection and vaccination-induced immunity into vaccination strategies and/or schedules may provide gains through the optimization of immunization schedules tailored to each country/community with different levels of community transmission. Likewise, the additional boosters' risk-benefit profiles for the previously infected individuals, as well as their optimal timing and dose, shall stay a matter of continuing interest. The increasing evidence from the literature have now been newly cited in the manuscript, as

below, in order to back up our statement.

page 5, lines 87–92

Past SARS-CoV-2 [...] (1). Individuals with a previous SARS-CoV-2 infection not only enjoy longer-lasting post-vaccination antibody levels, but also are known to achieve more efficient protection from clinical re-infection(2). For those with pauci-/asymptomatic infections who, for various reasons, did not receive an acute diagnosis, its later confirmation is only possible serologically(3,4). Therefore, robust serological assays are increasingly needed that assure reliable results despite of the waning antibody responses. ~~Thus, anti-nucleocapsid antibody decay becomes a special concern when policy makers are seeking for a reliable index (e.g. past SARS-CoV-2 infection) upon which variations in their immunization programs can be made, in order to prioritize the immunization of more vulnerable populations(2).~~

page 9, lines 209–232

Anti-nucleocapsid serology, assessed in parallel with the neutralizing antibody responses, shall guide us to improved understanding of our hybrid immunity against SARS-CoV-2. Indeed, no policies currently consider an individual's prior infection status nor serology in the designing of vaccination strategies. However, faced with the increasing evidence that vaccinees with a previous SARS-CoV-2 infection are better protected from the virus than those without(14,15), the additional boosters' risk-benefit profiles for the previously infected individuals, as well as their optimal timing and dose, shall stay a matter of continuing interest.

The reference list has been updated, according to their appearance in the text.

page 12, lines 282–286

4. Nakagama Y, Rodriguez-Funes MV, Dominguez R, Candray-Medina KS, Uemura N, Tshibangu-Kabamba E, Nitahara Y, Kaku N, Kaneko A, Kido Y. 2022. Cumulative seroprevalence among healthcare workers after the first wave of the COVID-19 pandemic in El Salvador, Central America. medRxiv 2022.02.06.22270565, Available at medRxiv:

12. Nakagama Y, [...]. J Clin Microbiol 60:e0226221.
13. Matsuura T, [...]. Kinetics of Anti-SARS-CoV-2 Antibody Titer in Healthy Adults Up to 6 Months after BNT162b2 Vaccination Measured by Two Immunoassays: A Prospective Cohort Study in Japan. JVAC-D-22-00871, Available at SSRN: <https://ssrn.com/abstract=4107500>.
14. Wei J, [...]; COVID-19 Infection Survey team. 2022 Antibody responses and correlates of protection in the general population after two doses of the ChAdOx1 or BNT162b2 vaccines. Nat Med 28:1072–1082.
15. Hall V, [...]; SIREN Study Group. 2022. Protection against SARS-CoV-2 after Covid-19 Vaccination and Previous Infection. N Engl J Med 386:1207–1220.
16. Huynen P, [...]. Long-term longitudinal evaluation of the prevalence of SARS-CoV-2 antibodies in healthcare and university workers. Sci Rep 12:5156.

Other editorial revisions:

Reference #6, originally from preprint is now published in EuroSurveillance. The reference list has been updated, as below.

6. Anda EE, Braaten T, Borch KB, Nøst TH, Chen SLF, Lukic M, Lund E, Forland F, Leon D, Winje BA, Kran AMB, Kalager M, Johansen FL, Sandanger TM. 2022. Seroprevalence of antibodies against SARS-CoV-2 in the adult population during the pre-vaccination period, Norway, winter 2020/21. Euro Surveill 27:2100376.26 March 2021. ~~Seroprevalence of antibodies against SARS-CoV-2 virus in the adult Norwegian population, winter 2020/2021: pre vaccination period. medRxiv <https://doi.org/10.1101/2021.03.23.21253730>~~

We realized to have failed in declaring one funding source; the Osaka City University Special Reserves Fund. We apologize for this non-reporting. It is a public fund, to which authors report no conflict of interest. *Funding* statement has been

updated.

page 11, lines 259–262

This research [...], ~~and~~—Osaka City University Strategic Research Grant [OCU-SRG2021_YR09] and the Osaka City University Special Reserves Fund for COVID-19.

June 28, 2022

Dr. Yasutoshi Kido
Osaka City University
Dept. of Parasitology and Research Center for Infectious Disease Sciences
Asahimachi1-4-3
Abeno-ku
Osaka 5458585
Japan

Re: Spectrum00986-22R1 (Detecting the waning serological response with commercial immunoassays; 18-month longitudinal follow-up of anti-SARS-CoV-2 nucleocapsid antibodies.)

Dear Dr. Yasutoshi Kido:

It is my pleasure to inform you that your manuscript has been accepted, and I am forwarding it to the ASM Journals Department for publication. You will be notified when your proofs are ready to be viewed.

Sincerely,

Eleanor Powell
Editor, Microbiology Spectrum
